# Cannabis Microbiome and the Role of Endophytes in Modulating the Production of Secondary Metabolites: An Overview

**DOI:** 10.3390/microorganisms8030355

**Published:** 2020-03-02

**Authors:** Meysam Taghinasab, Suha Jabaji

**Affiliations:** Plant Science Department, Faculty of Agricultural and Environmental Sciences, MacDonald Campus of McGill University, QC H9X 3V9, Canada; meysam.taghinasab@mcgill.ca

**Keywords:** *Cannabis sativa*, marijuana, hemp, microbiome, endophytes, secondary metabolites, Cannabinoids, gut microbiota, root microbiota

## Abstract

Plants, including cannabis (*Cannabis*
*sativa* subsp. *sativa*), host distinct beneficial microbial communities on and inside their tissues and organs, including seeds. They contribute to plant growth, facilitating mineral nutrient uptake, inducing defence resistance against pathogens, and modulating the production of plant secondary metabolites. Understanding the microbial partnerships with cannabis has the potential to affect the agricultural practices by improving plant fitness and the yield of cannabinoids. Little is known about this beneficial cannabis-microbe partnership, and the complex relationship between the endogenous microbes associated with various tissues of the plant, and the role that cannabis may play in supporting or enhancing them. This review will consider cannabis microbiota studies and the effects of endophytes on the elicitation of secondary metabolite production in cannabis plants. The review aims to shed light on the importance of the cannabis microbiome and how cannabinoid compound concentrations can be stimulated through symbiotic and/or mutualistic relationships with endophytes.

## 1. Introduction

Cannabis (*Cannabis sativa L.*) refers to genetically different biotypes of both (nonintoxicant) industrial hemp and marijuana [1]. Differentiating strains of hemp from marijuana is based on an arbitrary threshold point of the psychoactive compound, Δ9-tetrahydrocannabinol (THC) at 0.3%, a criterion established by Small and Cronquist [2].

Originating from the Himalayas, industrial hemp (*C. sativa* L.) is the most ancient domesticated crop. It is typically bred for seed and fiber, and also for multipurpose industrial uses such as oils and topical ointments, as well as fiber for clothing, and construction material for homes and for building electric car components [3,4]. Both hemp and herbal marijuana varieties are members of the *C. sativa* species; however, industrial hemp cultivars are cultivated for fiber products, edible seeds, and oilseed and nonpsychoactive medicinal drugs [1].

Herbal marijuana, a term designated for the form of cannabis that is used for medical and recreational purposes, produces some principal components of phytocannabinoids such as the intoxicating compound Δ9-tetrahydrocannabinol (THC), and with a therapeutic effect such as cannabinol (CBN), cannabidiol (CBD), cannabidiol-carboxylic acid, cannabigerol (CBG), cannabichromene (CBC), all of which are currently undergoing promising research [1]. In cannabis plants, cannabinoids accumulate as cannabinoid acids and nonenzymatically decarboxylized into their neutral forms during storage. The biosynthetic pathways of the major phytocannabinoids (CBC, CBD, CBG THC) with pentyl side chains-*C_5_H_1_*_1_ begins with the production of CBG which is produced by condensation of a phenol-derived olivetolic acid, a precursor of the polyketide biosynthetic pathway, and a terpene-based geranyl pyrophosphate, a precursor of the plastidal biosynthetic pathway. From CBG, Δ-THC, CBD, and CBC are synthesized each by a specific enzyme [1]. For more complete analyses of phytocannabinoid biosynthesis, see Andre et al. [5] and Hanus et al. [6]. Additionally, the noncannabinoid compounds, including terpenoids and flavonoids, deserve attention as they may provide anti-inflammatory activity [7].

Phytocannabinoids accumulate in all parts of the plant; however, they are more concentrated in specialized secretory structures, the trichomes of the female flower buds [5,8]. In addition to phytocannabinoids, cannabis produces a plethora of secondary metabolites that are produced as an adaptation for specific functions in plants mostly to improve plant growth or defence against biotic and abiotic stress [9]. These metabolites provide diverse biological activities for use in human medicine and the pharmaceutical industry [10,11]. The use of metabolic engineering approaches is promising as it opens up the possibility of increasing the production levels of desired targeted phytocannabinoid-derived compounds [10,11]. Interestingly, CBD exhibits strong antimicrobial properties against clinically relevant multidrug-resistant bacteria (MDR) such as the methicillin-resistant *Staphylococcus aureus* (MRSA) strains, and the drug-resistant *Mycobacterium tuberculosis* XDR-TB with minimum inhibitory concentration (MIC) ranging from 0.5–2 µg/mL. These activities compare favourably with standard antibiotics for these strains [12]. Essential oils of cannabis showed moderate potency with an IC_50_ of 33 µg/mL against several yeasts, including *Cryptococcus neoformans*, *Candida glabrata*, and *C. krusei* [13].

Before the legalized use of *C. sativa* in different countries, cultivation was restricted to hemp varieties of high-yielding fiber with significantly low levels of the psychoactive Δ9-THC. The recent legalization of cannabis in various countries, including Canada, Uruguay, and eleven states in the United States for the production of medical and/or recreational purposes, have generated demand not only for high yielding varieties of Δ9-THC and/or cannabinoids but firm and reliable cannabinoid profiles. However, the legality of cannabis for medical and recreational uses varies by country, in terms of its possession, distribution and cultivation, consumption and uses for medical conditions it can be used for [14,15].

Although beyond the scope of this review, it is worth mentioning the importance of the production methods and environmental conditions, all of which influence the production of commercial and high-grade medical and/or recreational marijuana under indoor cultivation [16]. The critical conditions for optimal cannabis growth, include light intensity, quality and photoperiod [17], storage temperatures and humidity [18], fertilization [19,20], abiotic elicitors including phytohormones [21,22], and the microbiome [23]. For cannabis, smaller quantities of the invisible ultraviolet (UV-B) light reportedly elicits Δ9-THC accumulation in leaves and buds [24,25], however, the effect of spectral composition on cannabinoid concentration remains tenuous. The stress response is one of the major factors that alter plant chemical composition [26]. Drought stress is known to reduce plant growth significantly but can also increase secondary metabolite content [27]. For cannabis and hemp plants, there is inconclusive evidence linking drought or decreased humidity to increased cannabinoid and Δ9-THC production [28,29]. More work is needed to understand better the role of water stress in cannabinoid and THC production.

This review aims to characterize the microbial diversity associated with hemp and marijuana, show with recent examples the diversity of microbial communities (endophytes) that internalize their tissues, and list the benefits that they confer to their hosts. We also highlight the values of the biologically active compounds produced by endophytes that contribute to increased plant fitness and tolerance against biotic and biotic stress. Moreover, we provide some evidence that the microbial bioactive compounds produced by some endophytes are derivatives and/or analogs of their associated host plants.

## 2. The Microbiome

The microbiome is a term that describes the collective genome of microbial communities, the so-called microbiota, which is associated with humans, animals, and plants. During recent years, the impact of microbial communities on shaping the host immune system and fitness of their host has gained attention [30]. The composition of microbiota residing in a host is affected by environmental conditions such as temperature, pH, and nutrient availability [31]. The overuse of xenobiotics in agriculture, along with the emergence of antibiotic and pesticide-resistance strains in agriculture and human medicine, can affect the host capacity to interact properly with the microbiota [30]. Compared to the number of studies on the microbiota of human subjects, there is a minimal number of studies focusing on economically agricultural crops. It is because the microbiota of agricultural organisms is affected by plant species and genotypes, developmental stages, root exudation, soil type, and environmental conditions.

Nevertheless, gut and root microbiota share commonalities concerning the regulation of host gene expression [32,33], enhancement of metabolic capacities of their hosts through catabolic genes [34,35], and suppression of harmful pathogens [36]. These are few illustrations of the commonalities between root and gut microbiota. The literature on this topic is found in recent reviews [37,38].

Strategic and applied research on the impact of microbial composition concerning human health recognizes the role of prebiotics that includes changes in the structure and diversity of the microbiota and stimulation of the activity of health-promoting bacteria such as *Lactobacillus* and *Bifidobacteria* [39,40]**.** One of the hot topics in gut microbiota is the nutritional strategy of adding dietary phytochemical compounds such as the secondary metabolites, flavonols, and quercetin, which can influence the immune function of the host physiology [41,42]. It is worthwhile mentioning that flavonoids and quercetin are important phytochemicals in cannabis, and their combination makes them potent antioxidants [7]. Data on the antioxidant potential of noncannabinoids are based on in vitro studies. Undoubtedly, their effects involving clinical trials deserve attention.

### The Plant Microbiome

Plants, including cannabis host distinct beneficial microbial communities on and inside their tissues, designated the plant microbiota from the moment that they are planted into the soil as seed. The plant microbiome is composed of specific microbial communities associated with the roots and the soil surrounding the roots (i.e., the rhizosphere), the air-plant interface (i.e., the phyllosphere), and the internal tissues of the plant, the so-called the endosphere [43,44]. Seeds harbour diverse groups of microbiota that are a source of bio-inoculum for juvenile plants promoting protection against biotic and abiotic stress at seed germination and later stages [45,46]. Vertical transmission of endophytes from seeds to seedlings occurs in rice, wheat, and bioenergy crops [47,48]. Each of these microhabitats provides suitable conditions for microbial life, which also has a respective function for the host. Plant microbiome is a contributing factor to plant health and productivity [49]. An increasing body of evidence highlights the importance of plant microbiome as a systemic booster of the plant immune system by priming accelerated activation of the defence system [50]. Many studies focused on the rhizosphere microbiome due to the soil-derived microbial diversity surrounding the root, and a potential source for selecting beneficial microbes that positively affect plant health [49,51,52]. Several reviews addressed the role of the rhizosphere microbiome in conferring disease suppressiveness and improving drought resistance [49,53,54]; others studied contributing chemical components to selective enrichment of microorganisms in the rhizosphere [55,56]. Generally, above-ground plant microbiota mostly originated from the soil, seed, and air adapt an endophytic lifestyle inhabiting tissues of the plant internally and play vital roles in plant development and fitness. These microbial communities that internally inhabit plant tissues, are referred to as endophytes, and play a crucial role in plant development and growth [57]. In this review, we use the term endophyte based on the definition of Petrini to signify ‘all organisms inhabiting plant organs that at some time in their life can colonize internal tissues without causing harm to their hosts [58].

## 3. The Functions of Plant Microbiome are Essential for the Host

There is a considerable amount of information on the functional role of microbial communities associated with plants and their internal tissues. Plant-growth promoting rhizobacteria (PGPR) and endophytes stimulate plant growth by producing phytohormones such as auxins [50] gibberellins (GAs) abscisic acid (ABA), and ethylene (ET), or by modulating the plant’s endogenous phytohormone levels [59,60]. Under greenhouse conditions, PGPR favoured plant growth and development, as well as plant secondary metabolites accumulation and, consequently, antioxidant capacity. Seed and root-exudated flavonoids are inducers for the nodulation genes in rhizobia-legume interactions, and in mycorrhization of host plants [61,62] which remarkably is comparable to the modulation of gut microbiota by dietary flavonoids.

In general, Proteobacteria, and especially γ-Proteobacteria, such as *Pseudomonas* and *Pantoea* are the dominant endophytic bacteria isolated from a variety of plant species [63]. Moreover, Gram–positive and Gram-negative bacteria, including *Pseudomonas, Azospirillum, Azotobacter, Streptomyces, Enterobacter*, *Alcaligenes, Arthrobacter, Burkholderia*, and *Bacillus* could enhance the plant growth and suppress phytopathogens [64]. Diverse strains of *Pseudomonas, Bacillus*, *Arthrobacter,* and *Pantoea* species associated with soybean and wheat roots exhibited growth-promotion properties such as phytohormone production, mineral solubilization, and the production of the enzyme 1-amino cyclopropnae-1-carbixylate (ACC) deaminase [65,66]. ACC deaminase reduces the endogenous level of the stress hormone ET by limiting the amount of plant ACC deaminase, and prevents ET-induced root growth inhibition. In return, it promotes plant growth and lowering stress susceptibility, in return, resulting in more nitrogen supply for bacteria [67].

As with bacterial endophytes, fungal endophytes can facilitate mineral nutrient uptake, promote plant growth and development, and induce defence resistance against pathogens [68,69]. Furthermore, they enhance abiotic stress tolerance, notably, the dark septate endophytic fungus, *Curvularia* sp. provided thermal protection for host plant at high temperature [69]. Indeed, bacteria and nonmycorrhizal fungi have the advantage of axenic propagation that places them as an ideal model of the agri-horticulture application.

One of the tools to control plant pathogens with the least impact on the environment is biocontrol. There are numerous examples of biocontrol activities of bacterial and fungal endophytes against pathogen invasion and diseases [43,70,71]. Various mechanisms underlie the beneficial effects of bacterial endophytes on their hosts. These include antibiotic production, induction of host defences, and immunity via induced systemic resistance (ISR), parasitism, competition, and quorum sensing [72]. Equally, endophytic fungi can protect plants against pathogens by triggering host resistance via systemic acquired resistance and ISR [73,74], or by antibiosis and mycoparasitism [71].

## 4. The Microbiome of Hemp and Marijuana

Understanding microbial partnerships with industrial hemp and medical and recreational marijuana can influence agricultural practices by improving plant fitness and production yield. Furthermore, marijuana and hemp are attractive models to explore plant–microbiome interactions as they produce numerous secondary metabolic compounds [75]. Together, the plant genome and the microbial genome inside plant tissues (i.e., the endorhiza) that forms the holobiont is now considered as one unit of selection in plant breeding, and also a contributor to ecological services of nutrient mineralization and delivery, protection from pests and diseases, and tolerance to abiotic stress [76]. Increasing evidence suggests that the host genotype influences the composition and function of certain critical microbial groups in the endorhiza, which, in turn, affects how the plant reacts to environmental stresses [45] with plant traits essential for hosting and supporting beneficial microbes. Particularly, populations of rhizospheric bacteria in disease suppressive soils are enriched and act as the first line of defence in the host plant against root pathogens, thereby activating secondary metabolite biosynthetic gene clusters that encode NRPSs and PKSs to enhance the level of defence metabolites [77]. A growing body of evidence signals that a two-step selection model where plant type and soil type are the main drivers of defining soil microbial community structure [78,79]. The soil type defines the composition of the rhizosphere and root inhabiting bacterial communities, whereas migration from the rhizosphere into the endorhiza tissue is dependent on plant genotype [80]. Accordingly, the influence of soil type and plant genotype on the microbial community structure of marijuana offers support of the two-tier system model whereby soil type is a determinant of microbial communities in the rhizosphere, and cannabis cultivars are a factor of community structure in the endorhiza [23]. This view that the rhizospheric microbiome influences the selection of the next generation cannabis cultivars that are resilient to biotic and abiotic types of stress opens up a new approach of breeding. Of particular interest, the community structure of endorhiza correlates significantly with cannabinoid concentration and composition [23]. Future studies on using microbial communities of cannabis not only increase fitness but augment derived metabolite production that are worth pursuing.

### 4.1. Fungal Endophytes Associated with Different Organs of Hemp and Marijuana

The diversity of fungal and bacterial endophytes associated with different tissues of hemp and marijuana sampled from various geographic and ecological regions is listed in Figure 1. Almost all of the nonsymbiotic fungal endophytes reported by several studies belonged to the Ascomycetes, except for two studies that reported the presence of strains belonging to the Basidiomycetes, such as *Irpex, Cryptococcus* [81] and *Schizophyllum commune* [82]. Depending on the geographical region, the abundance of fungal endophytes associated with cannabis tissues varied. For example, the abundance number of fungal strains belonging to *Aspergillus, Penicillium, Phoma, Rhizopus, Colletotrichum, Cladosporium, and Curvularia* in leaf samples from Himachal Pradesh, India [83] was higher as compared to those in stems and petioles [83]. Similarly, the fungal strains *Cochliobolus* and *Aureobasidium *isolated from Canadian hemp samples were abundant in leaf tissue [81]. Leaf, twig, and bud tissues of Bedrocan BV Medicinal marijuana from the Netherlands were associated with endophytic communities belonging to the *Penicillium* species (predominantly, *Penicillium copticola*), *Eupenicillium rubidurum, Chaetomium globosum,* and *Paecilomyces lilacinus* [84]. Different species of *Aspergillus* (*A. niger, A. flavus,* and *A. nidulans*), *Penicillium* (*P*. *chrysogenum* and *P. citrinum*), and some pathogens, such as *Rhizopus stolonifer, Alternaria alternata*, and *Cladosporium* sp. were found in marijuana stem tissues [83]. Moreover, strains belonging to *Alternaria, Cryptococcus, Aspergillus, Cladosporium,* and *Penicillium* [81,83,85] were isolated from marijuana and hemp petioles, whereas *Aureobasidium* and *Cladosporium* were isolated from hemp seeds [81]. Intensive mycorrhization of hemp roots by the arbuscular mycorrhizal (AM) fungi, *Diversispora* sp., *Funneliformis mosseae, Funneliformis geosporum*, *Glomus caledonium*, and *Glomus occultum* enabled the plant to tolerate soils contaminated with phosphogypsum and sewage sludge, and responded positively regarding biomass production [86]. It is highly probable that hemp selectively established relationships with mycorrhizal fungi to counteract abiotic stress through symbiosis.

### 4.2. Bacterial Endophytes Composition in Different Organs of C. sativa

The microbial community of bacterial endophytes associated with different cultivars of *C. sativa* belong to ϒ-proteobacteria and α-proteobacteria, including *Pseudomonadaceae*, *Oxalobacteraceae*, *Xanthomonadaceae*, and *Sphingobacteriales*, and all are well-known endophytic bacteria which substantiate observations from other plant systems (Figure 1) [87]. The most abundant strains isolated from leaves belong to *Pseudomonas* and *Bacillus*. Namely, *Bacillus licheniformis*, *Bacillus subtilis, Bacillus pumilus,* and *Bacillus megaterium* formed the most abundant Gram-positive bacterial endophytes population in the leaf [81,88]. Strains of *Pantoea* and *Staphylococcus* were associated exclusively with cannabis petioles [81], while strains of *Pantoea*, *Staphylococcus, Bacillus*, and *Enterobacter* were isolated from the seed [81]. The most prominent isolated genera from roots included *Acinetobacter, Chryseobacterium, Enterobacter, Microbacterium*, and *Pseudomonas* [87].

These findings prompted us to focus on whether cannabis-associated bacterial and fungal communities could (i) increase hemp and marijuana yield, (ii) control plant pathogens infection of cannabis plants, and promote disease resistance, (iii) modulate the production of cannabis secondary metabolites.

## 5. Endophytes, As Cannabis Microbial Biostimulants

Associated-bacterial endophytes with plant species can promote plant growth in plants via several mechanisms: Nitrogen fixation, siderophore production to chelate iron and make it available to plant roots, mineral solubilization mainly phosphorus and calcium, and production of several phytohormones including auxins, ABA, cytokinins, and GAs [75,79,89,90]. The production of such bioactive metabolites can enhance host plant growth and tolerate environmental stresses. There are limited studies on the use of growth-promoting bacterial endophytes and their effect on cannabis growth and yield. Pagnani et al. [91] evaluated the suitability of multispecies consortium consisting of *Azospirillum brasilense*, *Gluconacetobacter diazotrophicus*, *Herbaspirillum seropedicae*, and *Burkholderia ambifaria* isolated from roots or stems of corn, sorghum, sugarcane, and bermudagrass [92] to enhance hemp biomass. The bacterial consortium favoured plant growth development and the accumulation of secondary metabolites (i.e., CBD and THC). Conant et al. [93] reported on significant marijuana bud yield of 16.5% and plant height as a result of treatment with the microbial biostimulant Mammoth PTM, a multispecies consortium comprised of four bacterial taxa *Enterobacter cloacae*, *Citrobacter freundii*, *Pseudomonas putida,* and *Comamonas testosteroni* [94]. In the case of fungal endophytes, root inoculation of hemp by AM fungi enhanced tolerance of hemp to accumulate Cd, Ni, and Cr [95].

Most of the above findings illustrate the use of endophytic bacteria isolated from plant species other than hemp or marijuana with the ability to trigger some physiological plant responses. Our laboratory, along with other researchers, has reported on the diversity of endogenous fungal and bacterial endophytes and the abundance of taxonomic groups in different tissues of hemp and marijuana with growth promotion capabilities (Table 1) and biological control potential (Table 2) [81,82,83,84,87,96]. Some of these isolates were able to trigger the production of IAA-like molecules in the plant, reinforcing the notion that beneficial endophytes modulate plant development and growth through the production of phytohormones. However, the mechanism behind this is not fully clarified. Performing experiments with endophytes as growth elicitors would facilitate the evaluation of secondary metabolites profiles, particularly for THC, cannabinoids compounds, and terpenes of cannabis plants inoculated with endophytes.

Due to past legal restrictions on the production of marijuana and hemp, growth promotion trials applying endogenous microbiome isolated from hemp and marijuana are few. It seems reasonable to hypothesize that endogenous endophytic bacteria and fungi possess the genetic information to trigger phenotypic drastic growth promotion, and positively increase cannabis secondary metabolites in their respective hosts as compared to endophytes isolated from different plant species. With the legalization of marijuana in Canada and other countries, intensive investigations on how hormone-like molecules produced by endophytes influence plant adaptation and growth become possible.

## 6. Cannabis Endophytes with Antagonistic Effect Against Pathogens

There are limited bioprospecting studies on antagonistic activity of microbial endophytes associated with hemp and marijuana against invading pathogens and contaminating mycotoxigenic fungi [81,84,87]. These studies used the bioprospecting rationale that hemp and marijuana contain medicinal compounds that might also harbour competent microbial endophytes capable of providing health benefits to the host plant. The hemp-associated strains of *Pseudomonas fulva* (BTC6-3 and BTC8-1) and *Pseudomonas orientalis* (BTG8-5 and BT14-4), exhibited antifungal activities against *Botrytis cinerea* in dual confrontation assays [81]. These strains are top producers of hydrogen cyanide (HCN), cellulose, siderophore, IAA, and could solubilize P [81]. Additionally, *Pseudomonas* strains produce well-characterized secondary metabolites as diffusible antibiotics, including phenazines such as phenazine-1- carboxylic acid (PCA), 2,4-diacetylphloroglucinol (DAPG), pyocyanine, pyoluteorin, pyrrolnitrin, phloroglucinols, lipopeptides, and the volatile metabolite as HCN [99]. All these attributes make *Pseudomonas* strains effective biocontrol agents. The endophytic bacterial strains, *Bacillus megaterium* B4, *Brevibacillus borstelensis* B8, *Bacillus* sp. B11, and *Bacillus* sp. B3, employ quorum quenching as a strategy to disrupt cell-to-cell quorum sensing signals in the target organism [88]. This strategy provides defence against plant pathogens and prevents the pathogen from developing resistance against the bioactive secondary compounds produced by the plant and or the endophytes.

The cannabis endophytes, *Paecilomyces lilacinus* A3, *Penicillium* sp. T6, and *P. copticola* L3 successfully inhibited the growth of cannabis pathogens, *B. cinerea,* and *Trichothecium roseum* [84]. The endophytic strains of *Paenibacillus* sp. and *Pantoea vagans* successfully antagonized the pathogen *Fusarium oxysporum* in dual confrontation assays [87]. Taken together, these studies, although limited in scope, reveal the potency of endophytes in cannabis plants, and their applications hold great promise not only as biocontrol agents against the known and emerging phytopathogens of cannabis plants but also, as a sustainable resource of biologically active and novel secondary metabolites. These bioactive metabolites are an ideal substitute for chemo-pesticide not only to support low pesticide residue levels in cannabis flowers but also for adopting the zero-tolerance policy of pesticide residues in compliance with government regulatory bodies [100].

## 7. Endophytes of Medicinal Plants as Sources of Plants Secondary Metabolites

An exhaustive list of some of the same antimicrobial natural products biosynthesized by endophytes as their host plant is described in the recent review by Martinez-Klimova et al. [101]. The pharmaceutical molecules such as the antitumor drugs, vinblastine and vincristine [102], the anticancer drug camptothecin [103], the aneoplastic paclitaxel [104], and the insecticide azadirachtin [105] are amazing examples of the significance and importance of potentially valuable secondary metabolites produced by endophytes.

There is compelling evidence that both the plant and their endophytes can produce a collection of secondary metabolites from similar precursors, possibly as an adaptation of the host environment [106]. Some examples include podophyllotoxin [107,108], camptothecin, and structural analogs [103,109]. Some of these endophytes can biochemically produce compounds similar or identical to those produced by their host plants. It is proposed that such a molecular basis may attribute to horizontal gene recombination or transfer during the evolutionary process. For example, the ability of the taxol-producing fungus *Clasdosporium cladosporioides* MD2 associated with the host plant *Taxus media* is attributed to the gene 10-deacetylbaccatin-III-10-*O*-acetyl transferase. This gene plays a role in the biosynthetic pathway of taxol and bears a 99% resemblance to the host plant gene [106]. The latter endophytic fungus being the source of this important anticancer drug. The biosynthesis of the insecticide azadirachtin A and B by the fungal endophyte *Eupenicillium purvium* isolated from the Indian neem plant lends another evidence on the ability of endophytes to produce similar host plant metabolites [105]. The recent progress in the molecular biology of secondary compounds and the cloning of genes of endophytic metabolites offer insight into how the plant and endophyte genes of encoding the secondary metabolites are organized.

### 7.1. Endophytes Modulate Secondary Metabolites of Medicinal Plants

Accumulated evidence established that endophytes are capable of eliciting physiological plant responses, which in turn influence the production of secondary metabolites in the host plant [110]. The production of bioactive secondary metabolites of *Rumex gmelini* seedlings is enhanced through coculture with endophytic fungi [111]. An endophytic bacterium *Pseudonocardia* sp. induced artemisinin (antimalarial drugs) production in Artemisia plant [112]. Inoculation of the medicinal plant *Papaver somniferum* L. with a multispecies consortium increased the morphine yield by enhancing the expression of COR, an essential gene for morphine biosynthesis [113]. The alkaloid drug Huperzine A (HupA) used to treat Alzheimer’s disease is not only derived from the *Huperzia serrata* plant but also is produced and biosynthesized by the fungal endophyte *Penicillium* sp. LDL4.4 isolated from *H. serrata* [114]. In the legume Crotalaria (subfamily *Fabaceae*), the biosynthesis of pyrrolizidine alkaloids (antiherbivore, nematicide) depends on the nodulation by *Bradyrhizobium* sp. [115]. In another example, the bacterial and fungal endophytes associated with the Agarwood tree (*Aquilaria malaccensis*) enhanced the production of agarospirol, a highly sought after product in the pharmaceutical and perfumery industry, within three months of artificial infection [116]. Despite current research on the ability of endophytic microorganisms to produce plant-associated metabolites, their potential is not fully explored and is far from exhausted. Exploiting this complex plant-microbe relationship can only enhance the sustained production of phytochemicals by the associated microorganisms.

### 7.2. Possible Modulation of Cannabis Secondary Metabolite by Endophytes

Endophytes are well known to produce biologically active secondary metabolites that mimic the effect of the host plant metabolites or produce precursors of host plant compounds to activate the signaling pathway aiming to modulate secondary plant metabolites [117]. They induce the production of phytohormones such as ABA, GA, and ET that may provide a significant potential for improving cannabis secondary metabolites. Secondary metabolites, including THC, CBN, and CBD, are the most prevalent of cannabinoid compounds and inherently are employed in cannabis stress responses [118]. The precise role of cannabinoid in plant defence is not yet known. The plant growth regulators, including ABA, cycocel, ethephon, GAs, salicylic acid, γ-Aminobutyric acid (GABA), and mevinolin can manipulate cannabinoid biosynthesis and modulating secondary cannabis metabolites [22,118,119,120,121].

The potential for secondary metabolite recovery can be improved by the exogenous application of inducers. For example, the application of plant hormone GA_3_ at 100 µM level increased the amount of THC and CBD [120]. The exact mechanism of how the addition of exogenous hormones can affect the content of THC and CBD is not yet understood. One plausible hypothesis is that the exogenous application of GA_3_ contributes to the regulation of 1-aminocyclopropane-1-carboxylic acid [92] content, which in turn elevates ET levels that lead to higher THC and CBD contents [120]. Ethephon, another plant growth regulator, increased THC content of male flowers, and CBD content of female flowers [118]. Such an increase is attributed to ET levels that may function as a switch between growth and secondary metabolites synthesis. Accordingly, the exogenous application of two stress signaling molecules, salicylic acid (1 mM) and GABA (0.1 mM) improved THC content but deteriorated CBD content simultaneously. This effect suggests that these signaling molecules could affect the cannabinoid biosynthesis pathway through elicitation of expression of critical genes leading to eventual changes in the amount of the final products [22].

The concentration of cannabinoid compounds can be conceivably stimulated through biotic elicitation by symbiotic and or mutualistic relationships with endophytes. This raises the question of whether the production of identical molecules to plant hormones by endophytes in the plant would be useful as with the exogenous application of elicitors. A mixture of four bacterial endophytes significantly improved CBD and THC contents [91]. Endophyte could manipulate that ACC deaminase level, the precursor of THC biosynthesis in the plant [59,67,122]. Despite these advances, the mechanisms underlying the regulation of THC synthesis have not been completely elucidated.

It might be useful to draw an analogy between the medicinal plant-endophyte association and the engagement of the endophytes to produce structurally similar secondary metabolites of medicinal cannabis. However, the exact role of the natural products produced by endophytes inside cannabis from the perspective of helping in plant fitness is not precisely known. Unfortunately, this potential has not yet achieved.

## 8. Challenges and Future Directions

To date, basic information on cannabis endophytes diversity and composition is published. Most publications are restricted to isolation and identification of cannabis endophytes, but their biological effects on cannabis growth promotion and modulating of secondary compounds are unrevealed. Thus, it is imperative to understand the microbial partnerships with cannabis as it has the potential to affect agricultural practices by improving plant fitness and the production yield of cannabinoids. Interestingly, the active metabolites of microbial endophytes possess excellent biological activities that not only have the potential to wage war on plant biotic and abiotic stress, but are also useful for human health to prevent or cure fatal illness. The above observations highlight the wealth of untapped, and as of yet unknown functional traits of endophytes harbouring cannabis that need to be discovered and characterize their role in the enrichment of cannabis secondary metabolites. The importance of endophytic microorganisms producing compounds similar to their plants has gained momentum. Synthesized plant compounds by microbial endophytes are studied to produce secondary metabolites that are originally identified in their host plants. They could turn out some important medicinal compounds independently, which enable the pharmacological industry to large-scale fermentation of cannabinoids, independent of cannabis cultivation. This review emphasizes the great importance of more studies on cannabis endophytes and their biological properties. The examples presented in this review indicate that there is an urgent need to understand the molecular and biochemical mechanisms that might elicit similar responses in both plants and their associated endophytes that lead to the production of similar secondary metabolites.

## Figures and Tables

**Figure 1 microorganisms-08-00355-f001:**
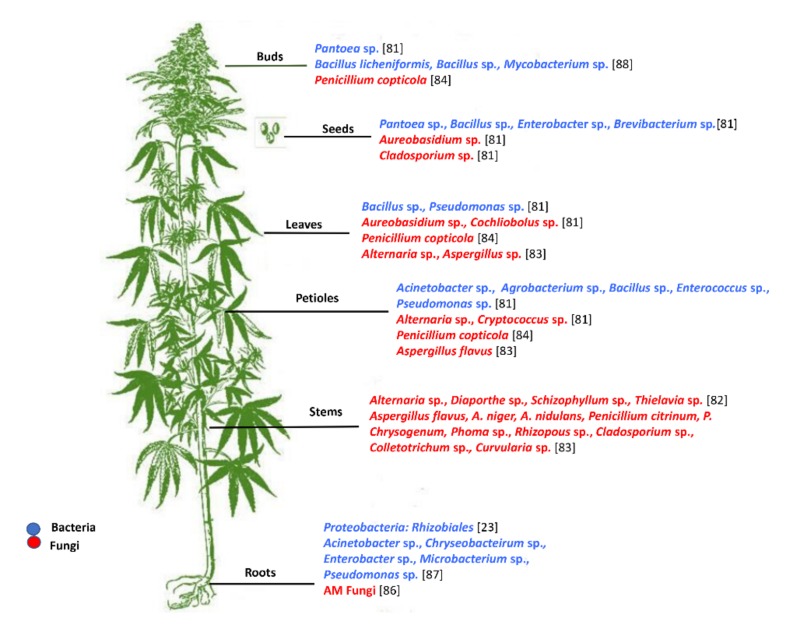
The most common endophytes harboured in different tissues of *Cannabis sativa* plants obtained from different geographical locations.

**Table 1 microorganisms-08-00355-t001:** Plant growth promoting bacteria and fungi associated with cannabis and their mode of action.

Organism	Activity	References
*Bacillus* sp.	P solubilizing	Joe et al. 2016 [97]
*B. amyloliquefaciens*	GAs production	Shahzad et al. 2016 [98]
*Pantoea vagans* MOSEL-t13	IAA production	Afzal et al. 2015 [87]
*Pseudomonas fulva* BTC6-3	P solubilizing and IAA	Scott et al. 2018 [81]
*P. geniculata* MOSEL-tnc1	IAA production	Afzal et al. 2015 [87]
*Serratia marcescens* MOSEL-w2	IAA production	Afzal et al. 2015 [87]
*Bipolaris* sp. CS-1	IAA and GAs production	Lubna et al. 2019 [96]

IAA: Indole acetic acid; Gas: Gibberellins; P: Phosphate.

**Table 2 microorganisms-08-00355-t002:** Cannabis endophytes with antagonistic effects against pathogens.

Organism	Target pathogen	Activity	References
Fungi			
*Penicillium copticola* L3	*Trichothecium roseum*	Growth inhibition	Kusari et al. 2013 [84]
*Paecilomyces lilacinus* A3 *Alternaria alternata* CN1 *Aspergillus niger* 2	*Botrytis cinerea* *Fusarium solani* *Curvularia lunata*	Growth inhibitionGrowth inhibitionGrowth inhibition	Kusari et al. 2013 [84] Qadri et al. 2013 [82] Gautam et al. 2013 [83]
Bacteria			
*Pseudomonas fulva* BTC8-1	*Botrytis cinerea*	Cellulase,HCN Siderophore	Scott et al. 2018 [81]
*P. orientalis* BTG8-5	*Botrytis cinerea*	Cellulase, IAA, Siderophore	Scott et al. 2018 [81]
*Paenibacillus sp.* MOSEL-w13	*Aspergillus niger* *Fusarium oxysporum*	Growth inhibition	Afzal et al. 2015 [87]

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
