# Peer review of "Cannabis Microbiome and the Role of Endophytes in Modulating the Production of Secondary Metabolites: An Overview"

_microorganisms, 2020, doi:10.3390/microorganisms8030355_

Round 1

Reviewer 1 Report

This is probably quite a timely little review of what is, from the perspective of animal biologists, a very misunderstood area of science, and potentially, is key going forward as the world becomes aware of the role of cannabinoid-based medicines. It should be published, but there are number of things the authors need to address first:

The grammar is not correct in many places, and the manuscript seems to have been written by more than one author – so it needs the attention of a really good English editor. For example, missing articles (such as “the”), pluralisation wrong in places and incorrect use of capitalisation. The font is wrong starting on line 173 The use of language is also a bit odd, for instance, in line 20 – what is meant by “cannabinoid compound concentration”, not really good English. Presumably it is the production that is being talked about. Clarify the term “elicitation” In the introduction, perhaps briefly outline how the cannabinoids are made in the plant, and the pathways involved. For instance, the plant generally produces the acid forms first. In the introduction, expand a bit on the known functions of the compounds in the plant, bearing in mind that most have evolved as part of a stress resistance paradigm. What is the role of UV and or drought in modulating their production? Provides a bit more information for the non-specialist reader. Provide some more history, cannabis has been used both as a medicine and a general crop for rope, oil etc., for 1000s of years, so that the review is a little more self-contained. Need to emphasise the point that both THC and CBD, and many other cannabinoids, are known to have anti-bacterial effects. This is an important clue. Which types of bacteria do they have the most effect on? What about concentration?

A key point about this paper is that it is very much written in the language of plant and prokaryote scientists, but there will be many readers who will be interested from the perspective of animal biology. So suggest introducing some comparisons to the rapidly evolving field around animal microbiota, of which a great deal more is known. The point here is that plants and animals both rely on their microbiota for survival. There is, for instance, an emerging interest in how plant secondary metabolites, such as the flavones, modulate gut microbiota in humans.

Author Response

Dear Madam/Sir,

Thanks for revising our paper and your suggestions have been immensely helpful.

1-We have corrected the grammar all throughout the review

2- We inserted the missing(example The) article wherever appropriate.

3- In introduction, lines 41 to 54 we briefly added how cannabinoids are made in the plant biosynthetic  pathways.

4- The role of the UV in modulating cannabinoid production was added on line 77 to 86, as well as the drought effect on lines 86 to 88.

5- More information in cannabis history in lines 35 to 40.

5- Cannabinoid concentration and its antibacterial effects were introduced more in lines 59-68

6- A new paragraph (section 2-The micobiome) is added to introduce the plant and gut microbiota  (see lines 99 to 123 ). WE also added on line 158 and onward the role of flavonoids in plant micrbiota

Reviewer 2 Report

This is a general review of endophytes and what is known of their ability to modulate secondary metabolites in plants. Where there is information the authors relate this to cannabis.

Throughout: cannabis, hemp and marijuana are common nouns and should not be capitalized. The authors need to take care and use the appropriate term cannabis, hemp or marijuana in the appropriate place.

The authors’ intentions are mostly clear throughout the text but they need the services of an editing/ proofreading service to bring their expression up to publication standard. I have worked on the first page (see attachment) but the rest of the manuscript needs a similar level of attention.

Line

9, 64       Change sentence as it implies that there is no seed transmission/association of endophytes.

18           only a modest part of the review focusses on cannabis

55           This review…

167         Fig 1: I think a more natural order would be buds, seeds, leaves, petioles, stems, roots

Author Response

Dear Madam/Sir,

Thanks a lot for review our manuscript and correcting the abstract. The authors appreciated your effort. The authors have taken extra precaution to correct the abstract.

1- In line 133 the vertical transmission of endophytes is added.

2- Figure 1 has been changed according to your comment.

Best regards,